# Enabling and constraining successful reablement: Individual and neighbourhood factors

**Christopher Justin Jacobi** [1]*, **Darren Thiel** [2], **Nick Allum**[2]

1 Department of Sociology, Nuffield College, University of Oxford, Oxford, Oxfordshire, United Kingdom,
2 Department of Sociology, University of Essex, Colchester, Essex, United Kingdom

* chris.jacobi@nuffield.ox.ac.uk

## Abstract

Using multilevel logistic regression to analyse management data of reablement episodes collected by Essex County Council, a UK local authority, this article identifies constraining and enabling factors for successful reablement. Overall, 59.5% of reablement clients were classed as able to care for themselves when assessed after 13 weeks following the reablement intervention (N = 8,118). Several age-related, disability, referral, and social factors were found to constrain reablement, but some of the largest constraining effects were neighbourhood deprivation as measured through the Index of Multiple Deprivation and, particularly, unfavourable geodemographic profiles as measured through Experian Mosaic consumer classifications. The results suggest that in order to optimise reablement, programmes should consider broader social and environmental influences on reablement rather than only individual and organisational aspects. Reablement might also be better tailored and intensified for client groups with particular underlying disabilities and for those displaying specific geodemographic characteristics.

**Data Availability Statement:** The dataset is available at the Open Science Foundation (DOI: 10.17605/OSF.IO/VEJAH) (URL: https://osf.io/vejah/).

**Funding:** Funding for initial analyses of the dataset was provided by an internal grant from the

## 1 Introduction

Generated by changing community structures and aging populations combined with government budget cuts and facilitated by 'empowerment orientated' treatment philosophies that aim to decrease dependence and increase self-sufficiency, short-term restorative health interventions involving physical and occupational therapy, health education, and/or assistive technologies, delivered outside of institutional settings and in clients' homes for limited time periods (for up to 6 weeks) have blossomed in health care provision in a number of Western nations, particularly in the UK [1–3]. Although the specific content of reablement programmes differs across and within countries, and also in relation to the particular needs of clients, all programmes share the aim to enable and 're-able' frail and disabled people to achieve 'functional independence', i.e. the ability to live a self-reliant life in which vital everyday activities like dressing, washing, eating, toileting and basic mobility are achieved by the clients themselves without the need for on-going assistance from homecare providers.

University of Essex under its Frontrunner Scheme (https://www1.essex.ac.uk/frontrunners/). The award (FR1213187) was jointly hosted and managed by DT (Department of Sociology). CJJ was selected as the recipient of the student funds of the award. All co-authors were all based at the University of Essex at the beginning of the project. CJJ's ESRC studentship grant (1368980) at the University of Oxford supported training in statistical methods. The funding bodies did not play any role in the study design, data collection and analysis, decision to publish or preparation of the manuscript. There was no additional external funding received for this study.

**Competing interests:** The authors have declared that no competing interests exist.

Reablement programmes may also forestall client admission to hospitals or other institutional care settings, possibly saving costs [3–7], and decreasing the probability of loss of functional independence following periods of hospitalisation [8, 9]. However, research evaluating reablement programmes has also demonstrated that substantial proportions of clients do not benefit from reablement any more than those that receive more traditional home care—despite their exposure to interventions that appear to be effective for others [9–11]. Moreover, systematic reviews of reablement tend to be inconclusive, showing rather mixed and contradictory outcomes [12, 13]. In this paper, we examine some of the factors that may contribute to the heterogeneity of outcomes, which have so far been little analysed in the reablement literature [13, 14], focusing in particular on the association between neighbourhood deprivation and the geodemographic profiles of reablement clients. To accomplish this, we use management data from Essex County Council, an English local authority, linked to geolocation information that contextualises clients in their local environments.

## 1.1 Reablement programmes

Compared to the provision of on-going homecare, reablement programmes have been shown by some studies (almost all of which are based on analyses of reablement for older people) to provide better outcomes for some clients in many areas including: subjective perceptions of quality of life [7, 11, 15–17] and mental health [18]; dementia [19–21]; increased independent coping with everyday activities [4, 17, 22, 23]; increased likelihood of remaining living at home rather than admission to hospital or institutional care [15, 17]; and a subsequent decrease in the levels and hours of on-going care provided by professional care workers [7, 11, 16, 17, 24, 25].

Despite some positive overall results, studies also show that not all clients become more independent following reablement, and that some clients tend to benefit more than others [7, 14]. We know little about the overall effectiveness of reablement, and even less about the types of clients that might benefit most, or least, from such programmes [3, 19], and there is scant evidence about broader socio-economic and neighbourhood factors that likely confound or facilitate successful reablement outcomes.

A handful of studies have identified some limited and limiting factors. Newbronner et al. [25] indicate that clients over 85 years old tend to benefit most from reablement, and Lewin et al [26] found that clients experiencing 'severe frailty' did not benefit from reablement intervention any more than those that received no intervention—yet people with mild or moderate frailties and those who lived alone tended to benefit most [see also 22]. In another study, Wilde and Glendinning [27] suggest that reablement was less successful for those with chronic disabilities and progressive conditions and also for those with sensory impairments, specific cultural needs, and, paradoxically in relation to Lewin et al., less effective for those that lived alone. There are clearly contradictions and knowledge gaps about which types of clients are more likely to benefit or not from reablement.

Part of these lacunae are a result of reablement being implemented, resourced and organised in different ways under different programmes [28–31], but another part is a result of the variability of clients in terms of their major disabilities and, as we demonstrate, their socio-economic status and local neighbourhood conditions. Identification of the influence of socio-economic and geo-environmental influences on reablement may provide an answer to some of the contradictory outcomes of reablement programmes–or, at least, raise important questions that have so far been largely excluded from the reablement debate.

Indeed, a number of recent studies have criticised reablement practice for its sole focus on individual clients without considering broader forms of informal support and homecare that

may be necessary to support independence–particularly over the long-term [31–34]. Studies indicate that successful reablement tends to dwindle over time [13, 14, 35], and it makes sense that, as people age, their needs change and reablement programmes thus need to be dynamic [29]. Yet it has also been posed that successful reablement and, long-term success in particular, are likely to be influenced by social conditions largely outside of an individual's control, but which are rarely considered in reablement policy or evaluation [31, 34, 36, 37]. These conditions include levels of informal social support for clients and broader neighbourhood and environmental conditions that may or may not aid independence. Neighbourhoods and local communities, alongside various socio-economic factors, are thus a central but missing ingredient in the reablement debate.

## 1.2 Neighbourhoods, health and reablement

Economists, public health researchers (gerontologists) and sociologists have for a long time emphasised the importance of neighbourhood conditions in shaping health [38–44], primarily through neighbourhoods' differing levels of social capital, physical environment, local services and stressors [45]. This may be especially likely for reablement because some of its major goals —like outdoor mobility (e.g. being able to walk in the local environment)—are directly linked to neighbourhood conditions [46]. Reablement success may thus be especially dependent on neighbourhood characteristics given that independent living necessarily requires access to local services like transport, doctors, chemists and shops, and because such conditions outside of the home will likely interact with people's independence in the home (e.g. cooking one's own meals would be dependent on access to shops to buy the ingredients). It is also likely, in this way, that informal social networks embedded in local neighbourhoods would be a key factor in supporting clients' recovery [47]. Moreover, as is the case for old people in general [46], reablement clients probably live in their respective neighbourhoods for long durations and, as they are less physically mobile, are especially vulnerable to local neighbourhood conditions. Low socio-economic status is also commonly associated with poor health outcomes [48–51], and some researchers have made effective use of measures that combine neighbourhood details and socio-economic status with a number of other measures, producing highly significant results [52].

To better understand the inconsistencies that we have shown to emerge from previous research, we use information about neighbourhoods and geodemography as predictors of relative reablement success. We do this by modelling detailed reablement management data retained by Essex County Council, combining this with commercial marketing data based on Experian's Mosaic classification data [53] and a more traditional measure of neighbourhood deprivation—the Index of Multiple Deprivation [54]. In contrast to more traditional measures of deprivation, Mosaic reflects affluence and consumption patterns which are used to categorise similar people into 66 distinct 'lifestyle types' and fifteen consumer groups (see Table 2 for descriptive statistics). Mosaic has been shown to bear associations with more traditional deprivation data [55], but it is also able to aid in the production of more detailed and fine-grained analyses and data [56]. The 'big data' provided through Mosaic presents new opportunities for researchers and policy analysts in the field of neighbourhood research [56], and here we contribute to this in our analysis of restorative care.

Exploiting these sources of geographically-based information, our principal research question is:

1. To what extent do neighbourhood-level factors influence the relative success rate of a reablement programme?

Successful reablement is defined as the share of reablement clients who are in 'self-care' i.e. the share of patients not requiring further care 13 weeks after the end of the reablement intervention. Relative success is defined as the association of predictors with reduced or increased probabilities of successful reablement in comparison to the overall rate of reablement success.

## 2 Data and methods

The data come from Essex County Council who monitored reablement programmes that were delivered by a specialist private care company (Essex Cares). The dataset contains 10,724 reablement client cases and represents the entirety of reablement programmes that took place in Essex between January 2008 to January 2012. Although the data is a little old and represents an early roll-out of reablement in the area, it provides a uniquely rich insight into reablement programmes by including the total population of reablement clients in Essex, classifications of their major disabilities and frailties, routes of referral to reablement, socio-demographic information and detailed geodemographic data.

In Essex, clients' care-needs were assessed by the care company through a Service Measurement Tool (SMT; see S1 Table) that measured clients' scores for mobility and transfers, ability for personal care, home skills, sensory abilities, levels of disability and cognitive understanding (largely mirroring Activities of Daily Living). Other categories in the assessment included the ability for communication, cooperation, and management of finances (largely resembling Instrumental Activities of Daily Living), as well as information about the levels of care currently received. As shown in S1 Table, each measurement was scored on a scale from zero (lowest level of independence) to four (full independence). Individual clients were then assigned a reablement package tailored to meet their individual needs as ascertained by the care company's use of the SMT.

Although the management data contained no information about the specific content of the reablement programmes delivered, all programmes were, in line with reablement philosophy, ostensibly tailored to individual client need. However, each programme commonly provided interventions in clients' homes for up to six weeks that aimed to teach the skills necessary to carry out everyday living activities in order to live independently or, at least, be less reliant on ongoing homecare services.

### 2.1 Case selection

As is typical with administrative datasets, various checks for plausibility (such as inconsistent or false codings, duplicate cases, and outliers) were conducted. All data management procedures were carried out in Stata 15.1 (StataCorp, College Station, USA). An overview of the data cleaning and case selection process is presented in Fig 1.

The chart shows that 204 cases were excluded because of coding errors and implausible values and that 1,524 so called repeat cases of patients who had already received a previous reablement intervention were also excluded because this paper focuses only on first-time reablement episodes. Due to data limitations on relevant covariates, we also restricted the analytic sample to people aged 60 to 99. Only 5.3 per cent (457 cases) of the remaining sample had to be dropped due to incomplete/missing information (listwise deletion). Robustness checks and multiple imputations for variables with missing values were carried out but they did not appear to change the results in any appreciable way.

The final dataset includes 8,118 clients with sufficient information at the start of the reablement programme. Clients were referred for reablement through either a stay in hospital or following referral by a care visitor. The data show that 1,454 (17.9%) clients were referred to reablement from a community context by a care visitor, and 6,664 (82.1%) from hospital. At

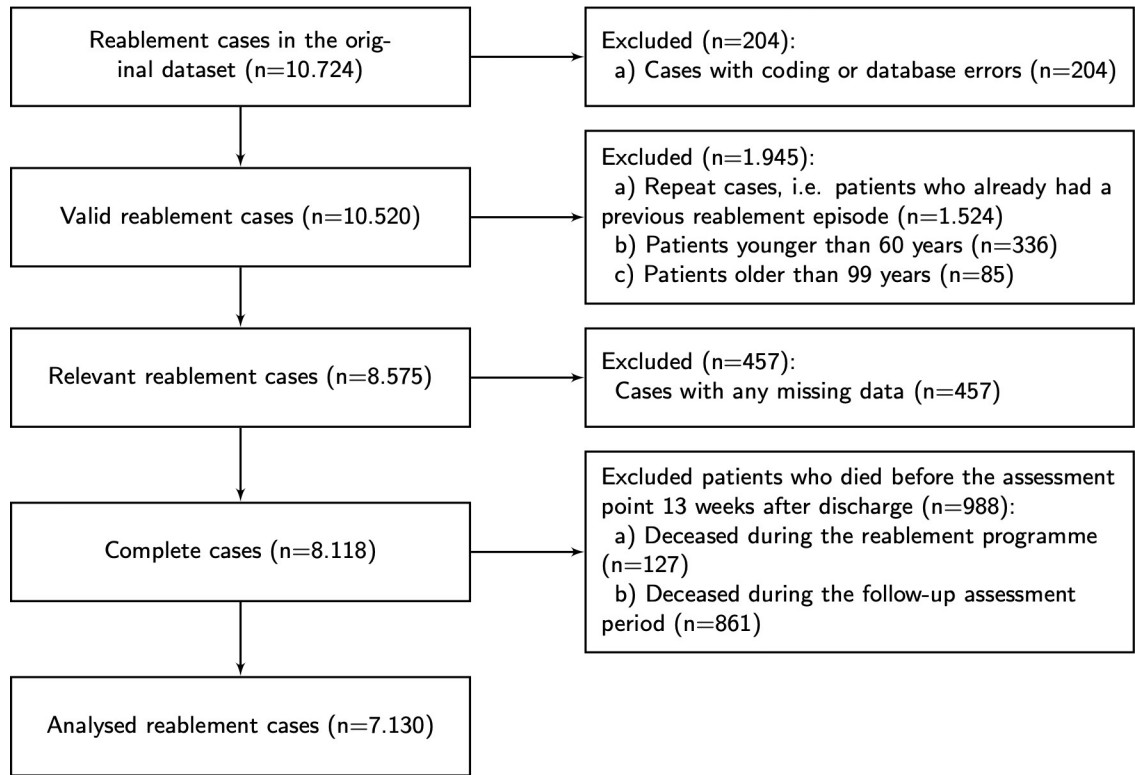

**Fig 1. Flow chart of case selection and data cleaning.** Administrative dataset of reablement episodes that were managed by Essex County Council, a UK local authority, from 01.2008 to 01.2012.

the time of assessment, that is 13 weeks after their discharge from the reablement intervention, 988 patients were deceased: It is highly unlikely that the reablement programmes would have prevented death over this short period of time, so these deceased patients have also been excluded in our statistical models. The selected reablement episodes of 7,130 patients had an average duration of 35 days and a median duration 38 days.

## 2.2 Variables

The outcome variable of our study is the ability to self-care 13 weeks after the reablement programme versus those clients that continued to need care, either as residential or homecare or in hospitals. Self-care at 13 weeks was used as the default follow-up period by the local authority and can be seen to represent a short to medium term assessment point of the effectiveness of the intervention. Even though patients were also assessed at discharge from the reablement programme, and at 26 and 52 weeks, these data points could not be used for further modelling because that data had been measured and coded inconsistently. Consequently, our modelling focuses on indicators of self-care at 13 weeks and does not include longer-term outcomes.

With the exception of the IMD scores, predictors in our models are categorical variables describing clients' general social characteristics (age, sex, ethnicity and marital status), referral route (hospital or community), two categorisations of clients' general levels of disability ('care-need' and 'care condition'), and geodemographic profiles using the Mosaic classification data. There were more women than men in the sample but this reflects the demographics of old age in the UK.

Four marital status categories were selected because the literature frequently states that married or cohabiting people tend to have different health and care outcomes than single people [57–59], and that cohabitation is likely to aid independence. Despite the relatively small number of the non-white ethnic group (N = 262), ethnicity (classified as white British or not) was also selected for analysis.

The coding of general levels of disability ('care condition') used at the initial assessment contained the following impairments: dementia, frailty, function, sensory, physical disability severe, physical disability appreciable, physical disability mild, and temporary illness. In addition, estimated daily social care-need was categorised into groups of 1–3, 4–6, 7–9, 10–12, 13–15 and 16–23 hours. The distribution of these variables is illustrated in Table 1 below.

Neighbourhoods were classified via the fine-grained Lower Layer Super Output Areas (LSOAs) based on the 2001 Census boundaries, with 866 separate areas in Essex which each represent, on average, 1,500 people. The LSOAs are designed to represent coherent geographic areas based on factors such as natural boundaries (rivers or roads) or population distributions [56]. To maximise the information provided by the dataset, we employed multilevel logistic regression models in which reablement clients were nested in LSOAs. The multilevel framework allowed us to capture the amount of shared variance in reablement outcomes at the neighbourhood levels and to adjust standard errors taking account of the hierarchically clustered data structure. The dataset contained at least one reablement case in 846 out of the 866 LSOAs in Essex, thus representing an almost complete neighbourhood coverage of 97.7 per cent. Further details can be found in the results section.

The LSOAs were matched with location data pertaining to levels of deprivation as measured by the Index of Multiple Deprivation (IMD) for the year 2007. The IMD, which is scored from 0 (least deprived) to 100 (most deprived), is a composite area-based indicator of income, employment, health deprivation and disability, education skills and training, barriers to housing and services, crime, and living environment statistics based on government data that is commonly used in UK social research [54]. A supplementary indicator of income deprivation affecting older people (IDAOPI) and two sub-indicators, each representing barriers to housing and services (i.e. geographic and wider barriers) and the living environment (i.e. indoors and outdoors), were also used. The IMD scores of each neighbourhood were double standardised so that a one-unit increase approximates the difference between one of the most to one of the least deprived neighbourhoods [60]. For purposes of mapping, a decile ranking of the overall IMD indicator was calculated based on its percentile scores, and Geographic Information Systems (GIS) techniques were employed to visualise the results via the user-written Stata command 'maptile' [61].

We also utilised the commercially-produced Mosaic geodemographic classification as a predictor of relative reablement success (Table 2). These data are derived from detailed demographic, financial, socio-economic and consumption data, as well as location, property value and property characteristic information [62, 63]. The data is gathered from broad sources including the UK census and council tax bands, and also proprietary datasets pertaining to property valuations, house sale prices, self-reported lifestyle surveys, a survey of adults' consumption of products, brands and media; and intelligence gathered through monitoring internet use [64]. Mosaic classifications can be linked with readily available location data (drawn by postcodes) in administrative datasets, which is particularly useful when socio-economic factors have not been directly collected. We had access to Mosaic data at the postcode level of the reablement clients (about 20 properties per postcode) for the year 2010 and employed them as detailed measures of the socio-economic conditions of the reablement clients. Of the 15 Mosaic groups, Group B ('Residents of small and mid-sized towns with strong local roots') was chosen as the reference category.

**Table 1. Distribution of individual-level predictor variables and their row percentages with the three reablement outcomes.**

| Variables | Continued care-need | Self-care | Deceased | Chi-squared |
|---|---|---|---|---|
| | Row % | Row % | Row % | Sig. |
| *Age groups* | | | | |
| 60–64 (n = 157) | 21.0 | 70.1 | 8.9 | 102.8 |
| 65–69 (n = 290) | 24.5 | 68.6 | 6.9 | *** |
| 70–74 (n = 434) | 26.5 | 64.1 | 9.4 | |
| 75–79 (n = 987) | 24.7 | 64.7 | 10.5 | |
| 80–84 (n = 1,727) | 25.7 | 63.4 | 10.9 | |
| 85–89 (n = 2,241) | 28.3 | 57.7 | 14.0 | |
| 90–94 (n = 1,690) | 32.7 | 54.9 | 12.4 | |
| 95–99 (n = 592) | 35.5 | 48.1 | 16.4 | |
| *Gender* | | | | |
| Female (n = 5,371) | 28.9 | 60.6 | 10.4 | 44.2 |
| Male (n = 2,747) | 27.2 | 57.2 | 15.5 | *** |
| *Ethnicity* | | | | |
| White (n = 7,856) | 28.4 | 59.3 | 12.2 | 2.5 |
| Non-white (n = 262) | 25.6 | 64.1 | 10.3 | . |
| *Marital Status* | | | | |
| Never married (n = 446) | 25.6 | 65.5 | 9.0 | 121.2 |
| Widowed (n = 4,037) | 32.4 | 56.7 | 11.0 | *** |
| Divorced/separated (n = 435) | 27.4 | 66.2 | 6.4 | |
| Married/cohabiting (n = 2,507) | 25.5 | 59.2 | 15.3 | |
| Missing (n = 693) | 17.7 | 68.7 | 13.6 | |
| *Referral route* | | | | |
| Community (n = 1,454) | 29.8 | 59.4 | 10.8 | 4.1 |
| Hospital (n = 6,664) | 28.0 | 59.5 | 12.5 | . |
| *Main care condition* | | | | |
| Dementia (n = 253) | 41.9 | 55.3 | 2.8 | 133.2 |
| Frailty (n = 2,115) | 32.6 | 57.2 | 10.2 | *** |
| Function (n = 85) | 30.6 | 65.9 | 3.5 | |
| Sensory (n = 113) | 23.9 | 59.3 | 16.8 | |
| Phys. dis. severe (n = 109) | 47.7 | 45.0 | 7.3 | |
| Phys. dis. apprec. (n = 4,741) | 26.6 | 59.9 | 13.5 | |
| Phys. dis. mild (n = 310) | 24.5 | 59.7 | 15.8 | |
| Temp. illness (n = 392) | 16.8 | 71.7 | 11.5 | |
| *Initial care needs in hours* | | | | |
| 1–3 (n = 215) | 25.6 | 60.9 | 13.5 | 141.2 |
| 4–6 (n = 2,303) | 24.1 | 64.9 | 11.0 | *** |
| 7–9 (n = 3,004) | 25.9 | 61.8 | 12.3 | |
| 10–12 (n = 1,689) | 33.7 | 54.4 | 11.8 | |
| 13–15 (n = 652) | 34.8 | 51.5 | 13.7 | |
| 16–23 (n = 255) | 45.9 | 35.7 | 18.4 | |
| Total (N = 8,118) | 28.4 | 59.5 | 12.2 | |

Administrative dataset of reablement episodes that were managed by Essex County Council, a UK local authority, from 01.2008 to 01.2012. Reablement outcomes were assessed 13 weeks after discharge from the reablement intervention.

**Table 2. Frequencies of the Mosaic groups per reablement outcomes.**

| Mosaic Group | Mosaic Group Description | Continued care-need | Self-care | Deceased | Chi-squared |
|---|---|---|---|---|---|
| | | Row % | Row % | Row % | Sig. |
| A (n = 293) | Residents of isolated rural communities | 27.6 | 63.1 | 9.2 | 78 |
| B (n = 1,631) | Residents of small and mid-sized town with strong local roots | 28.8 | 57.4 | 13.7 | *** |
| C (n = 116) | Wealthy people living in the most sought after neighbourhoods | 24.1 | 66.4 | 9.5 | |
| D (n = 769) | Successful professionals living in suburban or semi-rural homes | 24.6 | 62.8 | 12.6 | |
| E (n = 759) | Middle-income families living moderate suburban semis | 27.5 | 58.5 | 14.0 | |
| F (n = 224) | Couples with young children in comfortable modern housing | 20.1 | 67.4 | 12.5 | |
| G (n = 113) | Young, well-educated city dwellers | 24.8 | 64.6 | 10.6 | |
| H (n = 111) | Couples and young singles in small modern starter homes | 25.2 | 66.7 | 8.1 | |
| I (n = 216) | Lower income workers in urban terraces in often diverse areas | 34.3 | 49.5 | 16.2 | |
| J (n = 504) | Owner-occupiers in older-style housing in ex-industrial areas | 29.0 | 59.9 | 11.1 | |
| K (n = 579) | Residents with sufficient incomes in right-to-buy social housing | 33.9 | 54.6 | 11.6 | |
| L (n = 1,017) | Active elderly people living in pleasant retirement locations | 25.3 | 61.3 | 13.5 | |
| M (n = 1,632) | Elderly people reliant on state support | 30.8 | 59.7 | 9.5 | |
| N (n = 65) | Young people renting flats in high density social housing | 21.5 | 66.2 | 12.3 | |
| O (n = 89) | Families in low-rise social housing with high levels of benefit need | 39.3 | 42.7 | 18.0 | |
| Total (n = 8,118) | | 28.4 | 59.5 | 12.2 | |

Administrative dataset of reablement episodes that were managed by Essex County Council from 01.2008 to 01.2012. Repeated reablement episodes are excluded from the sample. Mosaic classifications 2010 (Experian Ltd.).

## 2.3 Statistical modelling

First, we present the outcome statistics of the reablement episodes when assessed 13 weeks after discharge from the reablement programmes. Second, we fit three multilevel logistic regression models to assess the effects of our predictors on the relative success rate of reablement: Model 1 is the baseline model that includes commonly used client-level information, model 2 adds the neighbourhood IMD deprivation indicator, and model 3 adds the Mosaic classifications. Additional to the IMD single indicator model, we also fit twelve alternative multilevel models, one separately for each IMD sub- or supplementary indicator, but only five of them (models 2a-e) were statistically significant. Furthermore, we tested for all two-way interaction terms between the predictors (e.g. care-need and sex) in all models, as well as cross-level interactions (e.g. between individual care-need and neighbourhood or Mosaic characteristics). None of these showed appreciable effects nor reached statistical significance, so these results are not presented.

To enhance the interpretation of the odds ratios of the multilevel logistic regression models, we present average marginal effects (AMEs) as they offer a convenient way to summarise the average change in the probability of the outcome ('successful reablement') for a one-unit increase in each of our covariates, estimated over all values of that covariate [65]. AMEs thus give an intuitive indication of the size of the effect over the full distribution of the independent variables. A 0.05 alpha (significance) level is used for the statistical tests and p-values, and 95% confidence intervals are provided in the graphs of the average marginal effects. Overall model fit is assessed via the AIC, Chi-squared and Log-Likelihood ratio tests. The intraclass correlation coefficient (ICC) is presented to describe the amount of within-cluster correlation of individual and neighbourhood-level reablement success rates.

## 2.4 Ethics

Before the dataset was made available to the researchers, Essex County Council conducted an internal review that did not identify any privacy or ethical concerns. The dataset has been fully anonymised. Ethical clearance was given in accordance with the regulations of the University of Essex, and the researchers took care to comply with all necessary data protection policies and practices.

# 3 Results

As seen in Table 3, the overall results reveal that after 13 weeks following the intervention, a total of 59.5 per cent of clients were classified as no longer requiring on-going care—i.e. they experienced 'successful reablement'. In contrast, 28.4 per cent of clients continued to need care and 12.2 per cent of clients were deceased. When excluding deceased clients, the 'success rate' of the reablement programme is 67.7 per cent—i.e. 4,828 out of 7,130 clients were in self-care 13 weeks after the programme.

Of course, we cannot know what proportion of clients would have achieved self-care regardless of receiving reablement or not. For instance, those referred from hospitals might be likely to move to increased self-care simply as result of healing over time due to natural recovery from physical injuries.

## 3.1 Model 1: Baseline multilevel model of client-level covariates

Fig 2 plots average marginal effects for each of the client-level covariates with associated 95 percent confidence intervals. The underlying multilevel logistic regression coefficients and statistical tests are shown in Table 4.

Looking first at age, compared to the reference group of people 85–89 years old, people slightly younger (80–84) had a four percentage point higher chance of successful reablement; people aged 90–94 a four percentage point worse chance; and people in the older age group of 95–99 years a nine percentage point worse chance of experiencing self-care at 13 weeks. The youngest age group—people aged 60–64—have an eight-percentage point higher chance of experiencing self-care than 85-89-year olds. The results imply that very old people have an especially reduced likelihood of self-care. The insignificant effect for the second and, marginally significant effect for third-youngest age groups ($p<0.1$), could stem from the fact there is much heterogeneity at younger ages and that some of them have a very severe care-need.

There is no statistically significant effect of ethnicity or sex. People referred from a hospital had a three-percentage point higher chance of experiencing self-care, though this effect only remained significant at the $p<0.1$ level in model three, and, as suggested above, it may be a

**Table 3. Reablement outcome statistics 13 weeks after discharge from the programme.**

| Reablement clients' outcomes 13 weeks after discharge from the programme | Statistics | |
|---|---|---|
| | **Count** | **Column %** |
| Self-care (i.e. successful reablement) | 4,828 | 59.5 |
| Continuing care-need (i.e. unsuccessful reablement) | 2,302 | 28.4 |
| Deceased | 988 | 12.2 |
| Total | 8,118 | 100 |

Administrative dataset of reablement episodes that were managed by Essex County Council, a UK local authority, from 01.2008 to 01.2012. The category of continuing care-need incorporates both homecare and residential care. The analysis is limited to first-time reablement episodes.

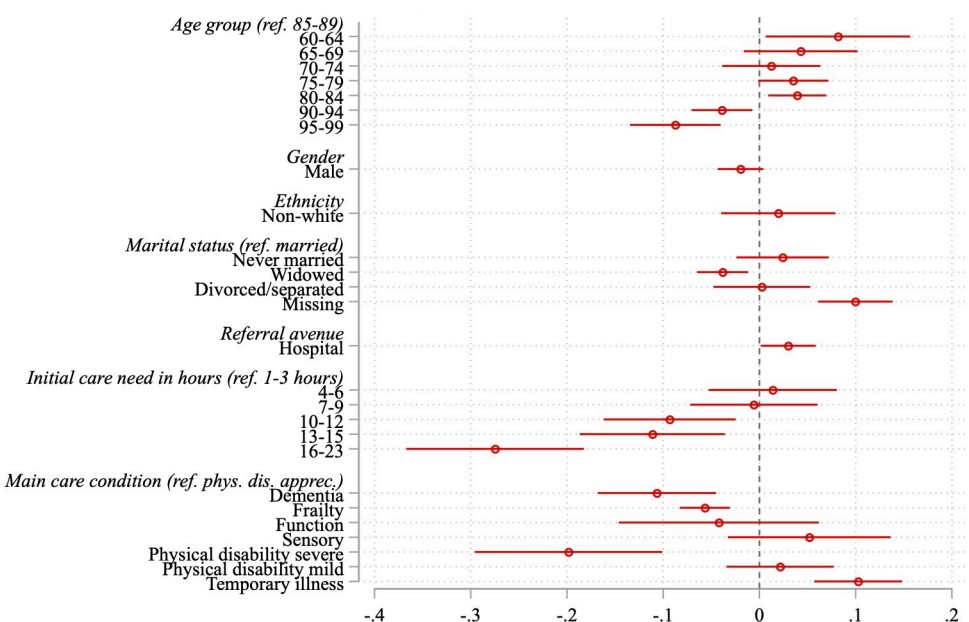

**Fig 2. AMEs of reablement success (self-care) on individual-level covariates.** Average marginal effects of the individual-level predictor variables (model 1) on the chances of self-care, 13 weeks after the reablement episode. Notes: Administrative dataset of reablement episodes that were managed by Essex County Council, a UK local authority, from 01.2008 to 01.2012. N = 7,130. Multilevel logistic regression model of reablement success (self-care 13 weeks after the end of the reablement episode) with individual-level covariates (age, sex, ethnicity, marital status, referral avenue, initial care needs in hours, and main care condition). Reablement clients (level 1) are nested in neighbourhoods (level 2) which are measured at Lower Layer Super Output Areas (LSOAs) in Essex, UK.

result of physical recovery rather than reablement. Compared to married reablement clients, those widowed had a four-percentage point lower chance of self-care. The missing category had a highly positive effect of 10 percentage points.

Predictably, when the amount of initial care-need in hours is higher, the chances of experiencing self-care at 13 weeks are lower. People with 13–15 hours of care-need have an 11-percentage point lower chance of self-care, and people in the highest category of 16 or more hours have a 27-percentage point reduced probability of self-care, which is a substantial difference. As seen initially, there is no significant difference for 1 to 9 hours of care-need, possibly because the hours of care-need are a broad measure and do not properly capture more detailed care needs. With regards to the main care condition, arguably a major reason for which reablement might be the solution, we see that people with frailty have a six percentage point lower chance of self-care at 13 weeks; those with dementia have an 11 percentage point reduced chance of self-care; and those with a severe physical disability have a 20 percentage point lower chance of self-care. People whose main condition is classified as a temporary illness have a 10-percentage point better chance of self-care. Functional, sensory and mild physical disability have statistically indistinguishable effects from appreciable physical disability (which is the comparison group among all conditions).

The intra-class correlations for models 1–3 reported in Table 4 are quite small at 0.02 (confidence interval: 0.01 to 0.04). Nonetheless, the Log-Likelihood ratio tests of the multilevel models compared to logistic regression models were highly significant at p<0.01 and hence confirm the need to account for the hierarchically nested structure of the data. Other model diagnostics, in terms of the AIC, show improved model fits when the neighbourhood

**Table 4. Multilevel logistic regression results of reablement success (self-care) after 13 weeks including IMD and MOSAIC categories (N = 7,130).**

| Variables | Model 1 | | Model 2 | | Model 3 | |
|---|---|---|---|---|---|---|
| *Age groups (ref. 85–89)* | | | | | | |
| 60–64 | 1.53* | (0.33) | 1.53* | (0.33) | 1.56* | (0.34) |
| 65–69 | 1.24 | (0.19) | 1.25 | (0.20) | 1.27 | (0.20) |
| 70–74 | 1.06 | (0.14) | 1.07 | (0.14) | 1.08 | (0.14) |
| 75–79 | 1.19+ | (0.11) | 1.19+ | (0.11) | 1.18+ | (0.11) |
| 80–84 | 1.22* | (0.09) | 1.21* | (0.09) | 1.22* | (0.09) |
| 90–94 | 0.83* | (0.06) | 0.83* | (0.06) | 0.83* | (0.06) |
| 95–99 | 0.67*** | (0.07) | 0.67*** | (0.07) | 0.66*** | (0.07) |
| *Gender* | | | | | | |
| Male | 0.91 | (0.05) | 0.91 | (0.05) | 0.91+ | (0.05) |
| *Ethnicity (ref. white)* | | | | | | |
| Non-white | 1.10 | (0.17) | 1.10 | (0.17) | 1.11 | (0.17) |
| *Marital status (ref. married)* | | | | | | |
| Never married | 1.13 | (0.14) | 1.13 | (0.14) | 1.14 | (0.14) |
| Widowed | 0.83** | (0.05) | 0.84** | (0.05) | 0.84** | (0.06) |
| Divorced/separated | 1.01 | (0.13) | 1.03 | (0.13) | 1.04 | (0.13) |
| Missing | 1.73*** | (0.20) | 1.73*** | (0.20) | 1.72*** | (0.20) |
| *Referral avenue (ref. community)* | | | | | | |
| Hospital | 1.15* | (0.08) | 1.16* | (0.08) | 1.15* | (0.08) |
| *Initial care needs in hours (ref. 1–3)* | | | | | | |
| 4–6 | 1.07 | (0.19) | 1.07 | (0.19) | 1.05 | (0.18) |
| 7–9 | 0.97 | (0.17) | 0.97 | (0.17) | 0.95 | (0.16) |
| 10–12 | 0.64* | (0.11) | 0.64* | (0.11) | 0.63** | (0.11) |
| 13–15 | 0.59** | (0.11) | 0.59** | (0.11) | 0.59** | (0.11) |
| 16–23 | 0.30*** | (0.07) | 0.30*** | (0.07) | 0.29*** | (0.06) |
| *Main care condition (ref. physical disability appreciable)* | | | | | | |
| Dementia | 0.61*** | (0.08) | 0.61*** | (0.08) | 0.61*** | (0.08) |
| Frailty | 0.76*** | (0.05) | 0.77*** | (0.05) | 0.77*** | (0.05) |
| Function | 0.82 | (0.20) | 0.82 | (0.20) | 0.82 | (0.20) |
| Sensory | 1.31 | (0.31) | 1.31 | (0.31) | 1.29 | (0.31) |
| Physical disability severe | 0.41*** | (0.09) | 0.41*** | (0.09) | 0.42*** | (0.09) |
| Physical disability mild | 1.11 | (0.16) | 1.12 | (0.16) | 1.12 | (0.16) |
| Temporary illness | 1.77*** | (0.26) | 1.78*** | (0.26) | 1.79*** | (0.26) |
| *IMD score (overall)* | | | 0.88* | (0.05) | | |
| *Mosaic (ref. B)* | | | | | | |
| A | | | | | 1.14 | (0.17) |
| C | | | | | 1.36 | (0.32) |
| D | | | | | 1.23+ | (0.13) |
| E | | | | | 1.05 | (0.11) |
| F | | | | | 1.61* | (0.30) |
| G | | | | | 1.22 | (0.30) |
| H | | | | | 1.33 | (0.32) |
| I | | | | | 0.69* | (0.12) |
| J | | | | | 1.07 | (0.13) |
| K | | | | | 0.81+ | (0.09) |
| L | | | | | 1.26* | (0.13) |
| M | | | | | 1.02 | (0.09) |

(*Continued*)

**Table 4.** (*Continued*)

| Variables | Model 1 | | Model 2 | | Model 3 | |
|---|---|---|---|---|---|---|
| N | | | | | 1.37 | (0.44) |
| O | | | | | 0.52* | (0.13) |
| Observations | 7130 | | 7130 | | 7130 | |
| Chi-squared | 299.8 | | 303.7 | | 335.1 | |
| AIC | 8691.2 | | 8687.9 | | 8679.3 | |
| ICC | 0.024 | | 0.023 | | 0.017 | |
| LR test (Chi-squared) | | | 5.24* | | 34.6*** | |

Exponentiated coefficients; Standard errors in parentheses

+ p<0.10

* p<0.05

** p<0.01

***p<0.001

Administrative dataset of reablement episodes that were managed by Essex County Council, a UK local authority, from 01.2008 to 01.2012. The results are based on multilevel logistic modelling of the chances of reablement success (self-care) on individual-level covariates, neighbourhood deprivation statistics at the LSOA level and the MOSAIC indicators (postcode level).

deprivation and Mosaic indicators are added into models 2 and 3, thus arguing further in favour of the multilevel analysis.

## 3.2 Model 2: Index of Multiple Deprivation (IMD)

In the analysed dataset (N = 7,130), on average, 8.4 reablement clients were living in each of the 846 LSOAs (minimum: 1; maximum: 37). As indicated in the methods section, the IMD was used to test if deprivation was associated with reablement outcomes and, while Essex is not a particularly deprived area of the UK overall, it contains some pockets of very high deprivation—most saliently seaside towns like Clacton, Harwich and Jaywick—the latter regularly being deemed the most deprived area of the UK [66–68].

Our results (Table 4) show that neighbourhood deprivation is a significant factor in reducing the relative likelihood of successful reablement—by 3 percentage points—an effect of a similar size as some of the individual-level disability predictors. The Log-Likelihood ratio test of model two versus model one is significant (Chi-squared = 5.24, p<0.05), suggesting an improved fit to the data. Furthermore, there are remarkable differences in the neighbourhood reablement success rates (mean 0.59; standard deviation 0.16), with some neighbourhoods having a 100 per cent success rate, and others a zero per cent success rate. However, the main part of this variability is associated with individual-level differences and the fact that despite the large size of the dataset, many neighbourhoods only have few reablement cases.

Because of this, we adjusted for the composition of reablement client-level characteristics within neighbourhoods through the multilevel specification of model one and we present the random effects estimates, i.e. the amount of neighbourhood variation that remains when accounting for the compositional effect of client-level variables. In Fig 3 we present two maps: the first for the IMD scores of the neighbourhoods; the second for the random effects estimates by the same LSOA neighbourhood boundaries. When comparing the two maps, there is not a perfect overlap between the IMD and the neighbourhood random effects estimates. For instance, many of the coastal areas around Jaywick in the north-east of Essex are extremely deprived but show moderately good reablement outcomes. Moreover, the neighbourhood success rate varies more in small clusters while the IMD shows a clear geographic pattern of pockets in the north-east (Clacton and Jaywick), south, and south-west (Harlow).

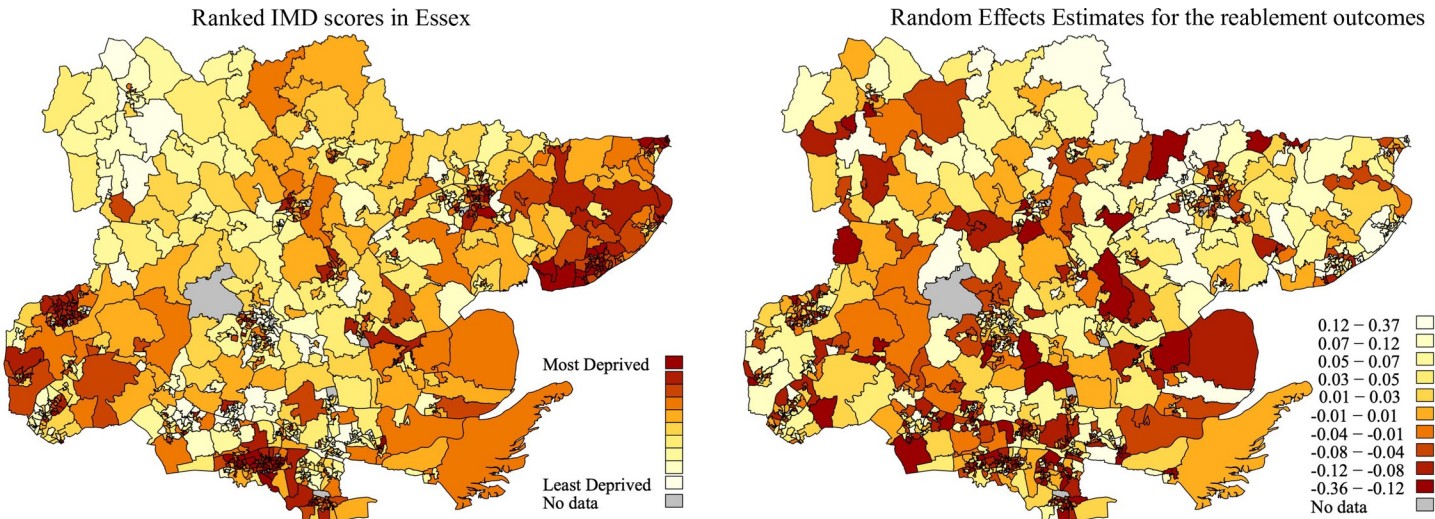

**Fig 3. Maps of neighbourhood deprivation (IMD) and random effects estimates of the reablement success (self-care) in Essex (N = 7,130).** 2011 Maps of the IMD scores per LSOA in Essex (A: left) and random effects estimates of the LSOAs in Essex (B: right). Notes: Administrative dataset of reablement episodes that were managed by Essex County Council, a UK local authority, from 01.2008 to 01.2012. A: IMD 2007 scores. LSOA of 2001 Census Boundaries. The Index of Multiple Deprivation (IMD) measures relative neighbourhood deprivation at the LSOA level. The IMD scores of the year 2007 are clustered into decile ranks from least to most deprived based on percentile scores. B: The random effects estimates are the shrunken residuals of the LSOAs after the multilevel modelling of self-care 13 weeks after the reablement intervention.

Compared to the IMD where some of the most deprived neighbourhoods are in densely populated areas, many of the neighbourhoods with below-average reablement success appear, by this measure, to be in medium or large areas (suburban/rural). Together, the results suggest that relative reablement success rates do differ between neighbourhoods. Some of this effect is captured via the IMD, but there is no direct overlap between the most deprived neighbourhoods and the best performing neighbourhoods in terms of reablement success rates. Administrative factors could play a role too, such as different reablement implementations between different local administrative divisions and hospitals.

Table 5 shows the multilevel models in which each of the seven individual components of the IMD, its four subdomains, and the supplementary indicator for income deprivation affecting older people (IDAOPI) are separately included. In terms of AME, the results reveal that only IDAOPI (-5 percentage points, p<0.001), crime (-4 percentage points, p<0.01), educational and skills (-4 percentage points, p<0.01), general income (-4 percentage points, p<0.01), and the sub-domain of the outdoor living environment—i.e. air quality and road traffic—(-3 percentage points, p = .05) reached statistical significance. It is noteworthy that neither the indoors deprivation subdomain nor the indicators for housing barriers are associated with the rate of relative reablement success, and that even the outdoor living environment is only marginally significant. These results suggest that general deprivation measured in terms of income and crime trump more specific indictors of neighbourhood deprivation that one might have expected to be closely related to reablement outcomes. This thus strengthens the notion of the importance of underlying socioeconomic factors on reablement. Nonetheless, because the overall IMD score incorporates data points of all deprivation measures, we focus the discussion on this overall measure.

### 3.3 Model 3: Mosaic geodemographic information

In order to distinguish the broad neighbourhood effects more granularly we utilised Mosaic geodemographic classifications at the postcode level. A Log-Likelihood ratio test of model

**Table 5. Alternative specifications (Models 2a-e) with the individual IMD indicators (N = 7,130).**

| Variables | Model 2a | | Model 2b | | Model 2c | | Model 2d | | Model 2e | |
|---|---|---|---|---|---|---|---|---|---|---|
| *Age groups (ref. 85–89)* | | | | | | | | | | |
| 60–64 | 1.52+ | (0.34) | 1.52+ | (0.34) | 1.53+ | (0.34) | 1.54+ | (0.34) | 1.52+ | (0.34) |
| 65–69 | 1.26 | (0.19) | 1.26 | (0.19) | 1.26 | (0.20) | 1.25 | (0.19) | 1.24 | (0.19) |
| 70–74 | 1.07 | (0.14) | 1.07 | (0.14) | 1.08 | (0.14) | 1.07 | (0.14) | 1.06 | (0.14) |
| 75–79 | 1.19+ | (0.12) | 1.18+ | (0.12) | 1.19+ | (0.12) | 1.19+ | (0.12) | 1.19+ | (0.12) |
| 80–84 | 1.21* | (0.09) | 1.20* | (0.09) | 1.22* | (0.09) | 1.21* | (0.09) | 1.21* | (0.09) |
| 90–94 | 0.83* | (0.06) | 0.83* | (0.06) | 0.83* | (0.06) | 0.83* | (0.06) | 0.83* | (0.06) |
| 95–99 | 0.67*** | (0.07) | 0.66*** | (0.07) | 0.67*** | (0.07) | 0.67*** | (0.07) | 0.67*** | (0.07) |
| *Gender* | | | | | | | | | | |
| Male | 0.91 | (0.05) | 0.91 | (0.05) | 0.91 | (0.05) | 0.91 | (0.05) | 0.91 | (0.05) |
| *Ethnicity (ref. white)* | | | | | | | | | | |
| Non-white | 1.11 | (0.17) | 1.10 | (0.17) | 1.09 | (0.17) | 1.12 | (0.17) | 1.10 | (0.17) |
| *Marital status (ref. married)* | | | | | | | | | | |
| Never married | 1.13 | (0.14) | 1.13 | (0.14) | 1.12 | (0.14) | 1.14 | (0.14) | 1.13 | (0.14) |
| Widowed | 0.84** | (0.06) | 0.85* | (0.06) | 0.84** | (0.05) | 0.84** | (0.06) | 0.84** | (0.05) |
| Divorced/separated | 1.03 | (0.13) | 1.05 | (0.13) | 1.03 | (0.13) | 1.03 | (0.13) | 1.02 | (0.13) |
| Missing | 1.73*** | (0.20) | 1.74*** | (0.20) | 1.73*** | (0.20) | 1.73*** | (0.20) | 1.73*** | (0.20) |
| *Referral avenue (ref. community)* | | | | | | | | | | |
| Hospital | 1.16* | (0.09) | 1.16* | (0.09) | 1.16* | (0.09) | 1.15+ | (0.09) | 1.16+ | (0.09) |
| *Initial care needs in hours (ref. 1–3)* | | | | | | | | | | |
| 4–6 | 1.07 | (0.20) | 1.07 | (0.20) | 1.07 | (0.20) | 1.06 | (0.20) | 1.07 | (0.20) |
| 7–9 | 0.97 | (0.18) | 0.97 | (0.18) | 0.97 | (0.18) | 0.96 | (0.18) | 0.97 | (0.18) |
| 10–12 | 0.64* | (0.12) | 0.64* | (0.12) | 0.64* | (0.12) | 0.64* | (0.12) | 0.64* | (0.12) |
| 13–15 | 0.59** | (0.12) | 0.59** | (0.12) | 0.59** | (0.12) | 0.59** | (0.12) | 0.60** | (0.12) |
| 16–23 | 0.29*** | (0.07) | 0.29*** | (0.07) | 0.30*** | (0.07) | 0.30*** | (0.07) | 0.30*** | (0.07) |
| *Main care condition (ref. phys. dis. apprec.)* | | | | | | | | | | |
| Dementia | 0.61*** | (0.09) | 0.61*** | (0.09) | 0.61*** | (0.09) | 0.61*** | (0.09) | 0.61*** | (0.09) |
| Frailty | 0.77*** | (0.05) | 0.77*** | (0.05) | 0.77*** | (0.05) | 0.77*** | (0.05) | 0.76*** | (0.04) |
| Function | 0.82 | (0.20) | 0.82 | (0.20) | 0.82 | (0.19) | 0.82 | (0.20) | 0.81 | (0.19) |
| Sensory | 1.31 | (0.31) | 1.32 | (0.31) | 1.32 | (0.31) | 1.32 | (0.31) | 1.33 | (0.31) |
| Phys. dis. severe | 0.41*** | (0.08) | 0.41*** | (0.08) | 0.41*** | (0.08) | 0.41*** | (0.08) | 0.41*** | (0.08) |
| Phys. dis. mild | 1.12 | (0.16) | 1.13 | (0.16) | 1.12 | (0.16) | 1.15 | (0.16) | 1.12 | (0.16) |
| Temp. illness | 1.78*** | (0.26) | 1.78*** | (0.26) | 1.78*** | (0.26) | 1.81*** | (0.27) | 1.80*** | (0.27) |
| *IMD sub-indicators* | | | | | | | | | | |
| Income | 0.85** | (0.05) | | | | | | | | |
| IDAOPI | | | 0.79*** | (0.04) | | | | | | |
| Education | | | | | 0.84** | (0.05) | | | | |
| Crime | | | | | | | 0.83*** | (0.05) | | |
| Outdoors | | | | | | | | | 0.86** | (0.05) |
| Observations | 7130 | | 7130 | | 7130 | | 7130 | | 7130 | |
| Log likelihood. | -4313.2 | | -4309.2 | | -4312.8 | | -4312.0 | | -4314.0 | |
| Chi-squared | 328.1 | | 335.6 | | 326.5 | | 332.4 | | 331.7 | |
| AIC | 8684.4 | | 8676.4 | | 8683.7 | | 8681.9 | | 8685.9 | |

Exponentiated coefficients; Standard errors in parentheses

+ p<0.10

* p<0.05

** p<0.01

***p<0.001

Administrative dataset of reablement episodes that were managed by Essex County Council, a UK local authority, from 01.2008 to 01.2012. All twelve sub- and supplementary Index of Multiple Deprivation indicators were individually modelled, but only the five presented ones reached statistical significance. The results are based on multilevel logistic modelling of the chances of reablement success (self-care) on individual-level covariates and neighbourhood deprivation statistics at the LSOA level.

three versus two confirms that the Mosaic improves the fit of the multilevel model to the data (Chi-squared = 34.65, p<0.01). The average marginal effects of the multilevel logistic regression of model 3 are shown in Table 4.

We find that compared to Mosaic group B ('Strong roots, mixed housing, small town, and tradition' characterised by 'Better off empty nesters in low density estates on town fringes'), clients in group O ('Disadvantaged, low income, long-term illness, low-rise council housing' characterised by 'Older tenants in low rise social housing estates where jobs are scarce') had a 14 percentage points lower chance of successful reablement (p<0.05); and clients in group I ('Few qualifications, ethnic diversity, small homes, crowded, below-average incomes' characterised by 'Older town centre terraces with transient, single populations') had an eight percentage points lower rate at (p<0.05).

On the other hand, a positive effect is seen for clients in group F ('Families, good incomes, comfortable homes, ethical products' characterised by 'Busy executives in townhouses in dormitory settlements') with an increased chance of nine percentage points (p<0.05). Group L ('Retired, bought a smaller property, specialist shops, grandchildren') had a five percentage points increased chance of successful reablement (p<0.05). Moreover, group D ('Significant equity, executives and managers, comfortable, good education, car ownership' characterised by 'Older people living in large houses in mature suburbs') showed a four percentage points increased chance of self-care at 13 weeks, but this effect was only marginally significant (p = 0.05).

The differences between the most negative and most positive Mosaic groups are relatively strong, often with as much or more influence on reablement success (self-care) than the major care condition predictors. Particular groups of affluent people enjoy significantly better reablement outcomes than the reference group (Mosaic group B), and those with less advantaged geodemographic profiles have worse outcomes. Additionally, the strongly negative effect for group O, the most deprived Mosaic group, shows that socio-economic disadvantage is reflected in a very high social care-need. It has been noted that social care services might also have communication challenges in effectively reaching these groups [69].

The significant positive effect for group F might stem from clients' proximities to their children, again stressing the importance of supportive social networks—in addition to factors like high incomes and comfortable housing. It is noteworthy that group L has a significant positive effect despite the high age of this group, suggesting that factors like purpose-built housing and functional environments have enabling influences at all ages. In general, advantaged socio-economic groups and people living in comfortable or purpose-built housing experience higher relative success rates of reablement and are thus more 'self'-reliant, while those in low-quality housing and those who are socio-economically disadvantaged tend to experience relative reablement success much less frequently.

## 4 Discussion

The nature of the monitoring data meant that there was no control or comparison group from which to analyse the overall effectiveness of reablement programmes versus other types of programmes of care in Essex, and there were no opportunities to interrogate the institutional processes underpinning the data collection or the ensuing quality of the data produced by the specialist care company. We therefore cannot ascertain the extent to which the effect of the reablement intervention is causal. Yet the findings do show the relative success of reablement as measured by the share of clients that were in self-care 13 weeks after the intervention, and they also highlight numerous client factors that had an influence on the relative rate of reablement success. Moreover, as a result of the reablement programme having been provided on a

large scale without strict selection criteria, it is unlikely that self-selection of comparatively healthier people into the programme affected our results. Importantly, it is the very large size of the dataset that makes our contextual (neighbourhood) and multilevel analysis possible.

The most influential predictive factors for negative reablement outcomes were related to previous care-needs where having very high hours of initial care-need, being much older, and having severe physical disability, dementia or frailty significantly mitigated the success of reablement. This confirms some previous research that high levels of disability are likely to have a negative effect on reablement success [5, 22, 26]. Similarly, our finding that people older than 90 were less likely to benefit from reablement is probably also the result of poor health and the natural limits on health improvements at an older age. This is not to say older people do not profit from reablement but it may be unrealistic to expect full self-care and, as studies have suggested, reablement for the very frail and disabled may often need to be accompanied by at least some level of home-care in order for it to be effective, especially over the long term [31, 33, 37]. Moreover, some level of informal care is likely to continue to aid independence for many reablement clients and, although this was not included in the management data and is thus an unknowable for our analysis, informal support has been shown by some studies [34, 37] to be an important component of successful reablement.

The outcomes of reablement for people with high levels of disability and thus high care needs require further research as it is probable that they are also contingent on the specific type of underlying disability and medical condition. Nonetheless, particular regressive health problems and disabilities mean that reablement is bound to be quite restricted for some groups. Similarly, it can be assumed that high initial care-need (as measured here by having care needs greater than 13 hours) will be unlikely to be wholly ameliorated through reablement and that some clients will continue to require some level of traditional home care alongside reablement [31, 33, 37].

Our analysis found that men did not have significantly different outcomes from reablement than women, and we did not observe any differences by ethnicity. However, only 3.2 per cent of the study population were non-white while 11.8 per cent of Essex residents in general are non-white, and this limits our conclusions. More significantly, we identified that hospital referral to reablement had a positive bearing on successful reablement. This could mean that cases referred from hospitals tend to be more acute and short-term, and thus more likely to be aided through reablement (e.g. simple improvements to the home environment following a fall), rather than what could potentially be more chronic and on-going health problems as in cases referred from the community.

Using the overall IMD score, we uncovered that living in a very deprived neighbourhood compared to a highly prosperous one reduced the chances of successful reablement by three percentage points. The largest effect (five percentage points) was reached when using the sub-indicator for income deprivation affecting older people. The second largest effect (four percentage points) was found when looking at the effect of the sub-indicator of crime in the neighbourhood. Generally, we found that there was non-trivial neighbourhood variation in the relative rate of reablement success and also that neighbourhood-level predictors were significantly associated with the likelihood of successful reablement.

Drilling further down into the geodemographic data, we found that specific Mosaic groups experienced very different reablement outcomes with, for example, group O ('Older tenants in low rise social housing estates where jobs are scarce') having a 15 percentage point lower chance of self-care after 13 weeks than the reference category; and group L (wealthy retirees) having a six percentage point higher chance. What this indicates is that the Mosaic categories associated with unpleasant living conditions (especially housing and broader environmental conditions), weak social networks and unfavourable socio-economic positions, had a highly

significant negative effect on reablement, while those indicating high socio-economic status had strong positive effects. Reassuringly in terms of our statistical modelling, the Mosaic groups that were not directly relevant for old people like group G ('young, well-educated city dwellers') were not significant, even though the direction of the effects followed the previously explored geodemographic direction. In a study with a similar design to ours, Nnoaham, Frater [70] used Mosaic geodemographic indicators and the IMD to predict uptake in colorectal cancer screening. They also found both measures to be statistically significant, with the Mosaic explaining a greater share of the variance [55].

Overall, the neighbourhood IMD and geodemographic Mosaic findings are evidence that there is a need to see successful reablement not merely as an individual or institutional process, but one intimately entwined with broader socio-economic, geographic, and community conditions. Indeed, as one recent reablement study explained [36]: 'The physical design of the community has a great impact on the seniors' ability to be active. Walking paths, benches, curbs, snow removal and icy roads are significant components. . . [Reablement] services cannot be integrated into the general health service without a focus on the municipality as a whole'. Our findings corroborate this, and they also demonstrate the relevance of the IMD and Mosaic measures for understanding reablement, and possibly as a more general tool for all community-based social care interventions. Following the 2014 English Social Care Act, reablement interventions are largely replacing traditional home care provisions and the presented findings on the role of neighbourhood conditions could therefore prove to be relevant for policymakers.

## 4.1 Limitations and further research

One caveat could be selective neighbourhood residence, namely a scenario in which clients more likely to benefit from reablement are more likely to live in better neighbourhoods. In neighbourhood research, this is often referred to as the issue of composition versus context [71]. Without longitudinal data of the clients' residence history or experimental evidence, this limitation cannot be ruled out. However, our results suggest clearly that local authorities could utilise Mosaic to further narrow specific characteristics such as housing conditions to better target and support reablement programmes, and that reablement, as also suggested by a number of recent studies [33, 34, 37, 72], should not be based on individual clients alone but should consider broader social and environmental conditions such as levels of informal help, local services and environments and adequate housing. Indeed, given that this study was based on the entire population of reablement episodes and was unusually large, the unprecedented level of detail is likely to represent a powerful insight into the role of neighbourhood factors.

In terms of avenues for further research, more studies are needed to see whether clients with multiple care-needs or so-called comorbidities have different reablement experiences. Moreover, in order to understand the significant factors that have been identified with more certainty and accuracy, further research is required into the specific nature of individual reablement programmes and the details of the underlying disabilities of reablement clients. Another issue is the long-term effectiveness of reablement programs—it might be that initial benefits are related to intervention effects where reablement initially boosts a client's positive outlook in the short-term but which become dampened over time when visits from reablement teams end and the realities of trying to cope every day on one's own sinks back into peoples' lives [27]. A possible way to mitigate against this could be to trial short 'booster' reablement sessions at regular intervals to these client groups [46], and to have some level of continued home-care.

Finally, alternative specifications could be used to measure reablement outcomes. This paper employed a strict test, namely full reablement towards self-care. While this approach carries the advantage of robustness compared to more relative outcome variables, future studies could also measure factors such as rates of improvement, maintenance, and deterioration of care needs, consider domain specific outcomes, and also employ more subjective evaluations based on quality-of-life indicators such as the EQ-5D-5L [19].

## 5 Conclusion

Using a uniquely comprehensive dataset provided by a UK local authority, this study highlights two major findings. Firstly, we illustrate the underappreciated importance of the socio-geographic environment for recovery and reablement–showing that neighbourhood level variables in terms of the Index of Multiple Deprivation are significantly associated with the chances of reablement success. Secondly, we show the utility of utilising commercial 'big data' like Mosaic geodemographic typologies for analytical purposes, which may uncover more detailed outcomes than traditional data [55, 56].

This has important implications for healthcare policy and practice, which should clearly consider broader socio-economic and social-geographic settings when planning, doing and assessing reablement. Moreover, rather than a consideration of only individual or organisational factors, future research into reablement might examine what these neighbourhood factors are so as to identify where reablement interventions might be of most benefit and which broader social-geographic factors are necessary for successful of reablement.

## Supporting information

**S1 Table. Reablement Service Measurement Tool (SMT).**
(PDF)

## Acknowledgments

We would like to acknowledge Essex County Council for the provision of the dataset. We are grateful for the academic input of Professor Berthold Lausen, Stephen Simpkin, Dr Duncan Wood, Vladimir Metodiev, Dr Ahmed Elhakeem, Dr Richard Hayhoe, and Danis Theodoulou. We also thank the reviewers of PLOS One for their extremely helpful comments and suggestions.

## Author Contributions

**Conceptualization:** Christopher Justin Jacobi, Darren Thiel, Nick Allum.

**Data curation:** Christopher Justin Jacobi, Darren Thiel, Nick Allum.

**Formal analysis:** Christopher Justin Jacobi.

**Investigation:** Christopher Justin Jacobi, Darren Thiel, Nick Allum.

**Methodology:** Christopher Justin Jacobi, Darren Thiel, Nick Allum.

**Project administration:** Christopher Justin Jacobi, Darren Thiel.

**Resources:** Darren Thiel.

**Software:** Christopher Justin Jacobi.

**Supervision:** Darren Thiel, Nick Allum.

**Validation:** Christopher Justin Jacobi, Darren Thiel, Nick Allum.

**Visualization:** Christopher Justin Jacobi.

**Writing – original draft:** Christopher Justin Jacobi, Darren Thiel.

**Writing – review & editing:** Christopher Justin Jacobi, Darren Thiel, Nick Allum.

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
