## [Decision Letter · Decision Letter 0]

25 Oct 2019

PONE-D-19-21555

Enabling and constraining successful reablement: Individual and neighbourhood factors

PLOS ONE

Dear Mr Jacobi,

Thank you for submitting your manuscript to PLOS ONE. After careful consideration, we feel that it has merit but does not fully meet PLOS ONE’s publication criteria as it currently stands. Therefore, we invite you to submit a revised version of the manuscript that addresses the points raised during the review process.

The manuscript has been reviewed by two reviewers. Both the reviewers have raised number of issues including the methods, lack of recent literature, structure of the paper and how the findings are relevant in the present context as the data used for the analysis were outdated. Hope the reviewers comments would be very useful to revise the manuscript.

We would appreciate receiving your revised manuscript by Dec 09 2019 11:59PM. To enhance the reproducibility of your results, we recommend that if applicable you deposit your laboratory protocols in protocols.io, where a protocol can be assigned its own identifier (DOI) such that it can be cited independently in the future. For instructions see: http://journals.plos.org/plosone/s/submission-guidelines#loc-laboratory-protocols

We look forward to receiving your revised manuscript.

Kind regards,

Kannan Navaneetham

Academic Editor

PLOS ONE

**Journal Requirements:**

**Comments to the Author**

1. Is the manuscript technically sound, and do the data support the conclusions?

Reviewer #1: Yes

Reviewer #2: Yes

2. Has the statistical analysis been performed appropriately and rigorously? 

Reviewer #1: I Don't Know

Reviewer #2: Yes

3. Have the authors made all data underlying the findings in their manuscript fully available?

Reviewer #1: No

Reviewer #2: Yes

4. Is the manuscript presented in an intelligible fashion and written in standard English?

Reviewer #1: Yes

Reviewer #2: Yes

5. Review Comments to the Author

Reviewer #1: Thank you for inviting me to review this manuscript which addresses an under-researched question in reablement/short-term social care interventions. I am not an expert in MLM so will not comment on the statistical components of this manuscript. Instead, I have drawn on my subject/methods expertise to provide topic and overarching comments.

Presentation:

Reablement is increasingly important internationally and the authors have drawn on international literature related to reablement, albeit not the most recent, relevant publications. However, they need to (re)consider some of the language/descriptors used (e.g. county council) and how these will be understood in the international context.

The authors use health-centric language throughout when describing both the intervention and the recipients of the intervention. However, both in the English and international contexts, reablement is not a medical intervention with eligibility determined by diagnosis but rather is a socio-therapeutic process/intervention with a focus on iADLs (not ADLs) where a person’s eligibility is determined by need and ability/capacity to learn/re-learn skills to manage life activities on a day-to-day basis, irrespective of diagnosis. Given this, the authors’ suggestion on page 19, for example, that further research is needed to understand which medical conditions are best tackled by reablement, is inappropriate.

The table and figure numbering is confusing.

The purpose of Figure 4 is unclear.

Methods:

Although it is acceptable for addressing the research question, as the data is quite old, further explanation is needed for the timescales of the data used and why only data up to 2012 are included (see comments below).

It was not clear in the manuscript, how the authors have addressed GDPR requirements in linking data from different sources - a statement about GDPR status would be useful.

There are some points of inconsistency: for example, between Table 4 and page 12. The lowest number of hours included in the 'initial care need in hours' category is 1-3 (as presented in Table 4). However on page 12, the authors state that that there is 'no significant difference from 0 to 9 hours'.

It is not clear at what point people's social care need had been assessed – was it at the point of entry into reablement? When they were in hospital? What they were already receiving from social care providers? Etc.? In short, is this the person’s assessed (i.e. their anticipated) need or their actual need? How long is the level of need expected to last?

Who undertook the assessment for initial care need in hours? The authors need to consider the implications of this in their discussion (e.g. potential for issues with estimating/recording need correctly?)

Thirteen weeks – is this 13-weeks post-assessment, or post-completion of the reablement intervention, or so on...?

Why was 13 weeks chosen as the appropriate follow-up timepoint?

Further explanation is required re: removal of repeat users of reablement.

Results/discussion sections:

The authors state that 59.5% of reablement recipients no longer required on-going care after 13 weeks. In the discussion, the authors need to consider whether this would have been expected irrespective of model of service provided (e.g. whether it is the reablement that made the difference, or for example, whether OT assessed provision of equipment/adaptations would have resulted in the same impact for the same types of people, etc.). How many of these would not have needed ongoing social care input whether or not they received reablement, irrespective of their local IMD status?

The data show that people with greater care needs received longer reablement episodes. The authors need to engage with this in the discussion section – does this result from reablement being used as a 'holding zone' for clients while awaiting a care package to be initiated, etc.

The levels of disability used by the council are a mixture of diagnoses and level of disability. Although the authors will not be able to improve the internal consistency of these categories, it would be helpful for them to consider the difference between a diagnosis and a physical state and amend the language used in this manuscript accordingly.

There is minimal critically engagement with the possible organisational (e.g. gaming re: data recording, goal-setting, etc.) and contextual explanations for results.

The authors need to explain how their data is relevant for today’s context. For example, eligibility, and therefore assessment of need, for social care has changed significantly since the introduction of the English Care Act in 2014. The authors need to critically engage with this change and explain how their findings are relevant for services assessing care need in hours today.

Reviewer #2: This is an interesting article looking at reablement services in Essex. There are a number of points that the authors should consider addressing.

-Generally, the article can do with a bit more succinct presentation of the results and methods.

-A lot of the information presented as supplemental materials should be included in the main text.

-Descriptive statistics could be split by categories of the outcome within the first table for all the baseline variables. Furthermore, those variables that were looked at baseline and followed up should be presented at both instances.

-Discuss whether the profile of those with missing values differ from those without on key variables.

-In some instances, information presented in the results section are more suited for the methods section, for example, when introducing AMEs, ICC, and AIC. Also, some methods are presented in the discussion where it should have been presented first in the methods, for example, interactions. Also, some information in the methods section could have been presented in the results section.

-Discuss the goodness-of-fit of the models presented.

-Model estimates should be included in tables in the main text include confidence intervals as well and the actual P-values.

Further description of the Mosiac variable regarding the 13 categories.

-How do the results compare to those of "Outcomes of reablement and their measurement: Findings

from an evaluation of English reablement services" https://onlinelibrary.wiley.com/doi/full/10.1111/hsc.12814. Also, the literature cited seem to be limited to 2017 or earlier, the authors might want to update their literature.

-Some discussion about the challenges of the data at hand and the restructuring of services.

-P-values that are very small should be reported as P-value < 0.001

-A flow diagram of the participants would be of help

-The data are quite dated some of the implications of this warrant a discussion.

-Level of statistical significance should be presented in the methods section.

-Statistical methods should be clearly marked as such in the methods section.

- Statistical significance should be clearly indicated across the manuscript, the authors do this in some instances but not all. Also remember absence of evidence is not evidence of absence.

6. PLOS authors have the option to publish the peer review history of their article (what does this mean?). If published, this will include your full peer review and any attached files.

Reviewer #1: No

Reviewer #2: No

---

## [Author Response · Author response to Decision Letter 0]

2 Feb 2020

We have responded to all the insightful and detailed reviewers’ queries about our paper, which we found very constructive. We think this has substantively strengthened the quality of the paper and helped us make a stronger case for our main findings. 

Both reviewers asked to update the literature: 

• We conducted a new literature search and uncovered 19 new papers on reablement which have integrated into our literature review and discussion sections – this can be seen throughout the paper.

• The newer literature helped to strengthen the relevance of geo-social factors – as can be seen in the new paragraph added to section 2.2.

• We have added a bibliography of all new papers (the 19 new ones on reablement and additional papers on the Mosaic indicators and methodology) at the end 

Reviewer #1:

Presentation: UK-based and heath-centric language 

• We have changed, throughout, the framing of our description of various factors related ‘frailty’ and ‘disability’ – rather than our previous use of the terms ‘health’ and ‘health condition’

• We have also changed the term ‘County Council’ to ‘local authority’ throughout the paper.

The table and figure numbering are confusing:

• We have double checked the numbering for consistency and against the guidelines, and as suggested by the reviewer we have moved more figures and tables into the main body of the text. 

• This was a very helpful suggestion and has added a lot of clarity. 

The purpose of Figure 4 is unclear. 

• We agree - this figure only added marginally new information and have thus removed it.

Methods: The data is old

• We have added a sentence at the beginning of section 3 (p. 7) that explains why this data set, although slightly old, is useful and helpful to understand reablement 

outcomes today. The unique size of the dataset means that we can accurately assess the role of neighbourhood effects – a new and innovate way of utilising ‘big data’ in reablement research. 

• We note that the theme of neighbourhood effects in reablement programmes is starting to take off at this present time, so our paper is extremely timely. 

It was not clear in the manuscript, how the authors have addressed GDPR requirements in linking data from different sources - a statement about GDPR status would be useful. (reviewer 2 also wants this)

• We have created a dedicated subsection on ethics and data protection in the methodology (section 3.4).

• The GDPR does not retrospectively apply to our data, and thus should not be a concern.

Mosaic: Need more explanations and details.

• We have added to and re-organised some of the description of Mosaic on P. 6 and P. 13, and we have also now referred to two 2019 studies about the use of Mosaic in health and ageing studies (Wami et al. 2019 and Moon et al. 2019) which provide additional detail about Mosaic categories and their resonance and relevance

There are some points of inconsistency: for example, between Table 4 and page 12. The lowest number of hours included in the 'initial care need in hours' category is 1-3 (as presented in Table 4). However on page 12, the authors state that that there is 'no significant difference from 0 to 9 hours'. 

• This was a copying error. Patients with less than one hour of care have now consistently been rounded up as needing one hour of care. 

It is not clear at what point people's social care need had been assessed – was it at the point of entry into reablement? When they were in hospital? What they were already receiving from social care providers? Etc.? 

• We state on p. 10 that our data had two routes to the Essex reablement – community and hospital but we have now made this more apparent with short discussion about the effects of referral routes on P. 16 and p. 25. We describe how some clients were already known to the Essex Care team and some already had a carer, while others were referred to reablement without any previous care history.

• Care need was assessed as the present need of care (P. 8).

Who undertook the assessment for initial care need in hours? The authors need to consider the implications of this in their discussion (e.g. potential for issues with estimating/recording need correctly?)

• The initial assessment was undertaken by Essex Cares (p.8)

• We have added a new sentence about these potential issues at the start of our discussion (p. 24).

Further explanation is required re: removal of repeat users of reablement.

• We have created a methodology subsection on case selection, and created a flow chart to clarify the case selection process (Fig 1). Again, we believe that this has greatly improved the clarity of this section and of the paper on the whole.

Thirteen weeks – is this 13-weeks post-assessment, or post-completion of the reablement intervention, or so on...? Why was 13 weeks chosen as the appropriate follow-up timepoint?

• We have now clarified this on p. 11 and in table 1. 

• 13-weeks is the default social care and reablement assessment point according to the data provider.

Results/discussion sections:

The authors state that 59.5% of reablement recipients no longer required on-going care after 13 weeks. In the discussion, the authors need to consider whether this would have been expected irrespective of model of service provided

• We have now clarified this issue throughout the paper and in the discussion. We also now clarify that our data does not allow us to make a causal claim (see pp. 24-25) 

The data show that people with greater care needs received longer reablement episodes. The authors need to engage with this in the discussion section – does this result from reablement being used as a 'holding zone' for clients while awaiting a care package to be initiated, etc. 

• We agree that the length of the reablement statistics were underdeveloped. We have thus removed them from table as they do not relate to the main outcome variable of this study.

• We have expanded the discussion of the interrogation possibilities of the data. We have targeted self-care at 13 weeks as the outcome variable because it is more objective than related improvement indicators.

The levels of disability used by the council are a mixture of diagnoses and level of disability. Although the authors will not be able to improve the internal consistency of these categories, it would be helpful for them to consider the difference between a diagnosis and a physical state and amend the language used in this manuscript accordingly.

• Similar to query as out use of health-centric language - we have now amended that throughout the paper as above.

There is minimal critically engagement with the possible organisational (e.g. gaming re: data recording, goal-setting, etc.) and contextual explanations for results.

• We have added a proviso about that at the start of our discussion on p. 24.

• We would also argue that our focus on neighbourhood effects is not as prone to bias from these institutional factors as in other studies that only have patient-level data. These external data should strengthen the robustness.

The authors need to explain how their data is relevant for today’s context. For example, eligibility, and therefore assessment of need, for social care has changed significantly since the introduction of the English Care Act in 2014. The authors need to critically engage with this change and explain how their findings are relevant for services assessing care need in hours today.

o We have read the 2014 English Care Act and added a footnote about it. We feel that this makes out paper even more relevant.

o We explain that the underlying principle of reablement, which is always targeted individually and varies by local authority in any case, can accurately be addressed with our unique dataset.

Reviewer #2: (new points not addressed above)

In some instances, information presented in the results section are more suited for the methods section, for example, when introducing AMEs, ICC, and AIC. Also, some methods are presented in the discussion where it should have been presented first in the methods, for example, interactions. Also, some information in the methods section could have been presented in the results section.

• This was a very helpful comment. We have moved several small paragraphs around and rewritten sentences so that the organisation of the materials flows much more naturally and is clearer for the reader. All the descriptive information about the data and variables is now in the methodology which now benefits from dedicated subsections.

• We have expanded the presentation of the statistical techniques in the methodology so that we don’t need to reference them individually in the results section.

A lot of the information presented as supplemental materials should be included in the main text.

• This is a very helpful comment. We were too strict initially and have moved three figures into the main body of the text.

Descriptive statistics could be split by categories of the outcome within the first table for all the baseline variables. Furthermore, those variables that were looked at baseline and followed up should be presented at both instances.

• We appreciate the comment, but given the other changes that have been carried out this was no longer needed. As this is not a longitudinal study, the covariate effects are best illustrated through the regression tables and charts. 

Discuss whether the profile of those with missing values differ from those without on key variables.

• The added flow diagram (Fig 1) provides a much clearer overview of the data selection process and the amount of missingness in the dataset. With only 457 out of an original 10,724 cases (4.3%) this is a very small amount, especially for a social science dataset of sensitive data on health status and neighbourhoods.

• Nonetheless, we conducted exploratory analyses of too see if the samples with and without missingness differed substantially on the covariates, but we identified only negligble patterns.

• The covariate imbalance plot below demonstrates that even in the most extreme case, for the sub-indicators of health conditions (function and temporary illness), there is only a 20% standardizes difference with respect to the reablement outcome. Again, this did not change the results

Level of statistical significance should be presented in the methods section. Statistical methods should be clearly marked as such in the methods section. Statistical significance should be clearly indicated across the manuscript. P-values that are very small should be reported as P-value < 0.001

• This was a good suggestion and we have implemented this change throughout the paper.

A flow diagram of the participants would be of help

• We have added the flow diagram (Fig 1); it has greatly improved the paper.

Discuss the goodness-of-fit of the models presented.

• We now present and discuss the goodness-of-fit measures numerous times in the paper and relate it the purpose of multilevel modelling (e.g. pp. 10 (Table 1), 13, 17, 19).

Model estimates should be included in tables in the main text include confidence intervals as well and the actual P-values.

• Point of confusion of AMEs and logistic regressions which are otherwise the same

Generally, the article can do with a bit more succinct presentation of the results and methods.

• We have significantly edited and reorganised the results and methodology sections: The paper is much more readable and clearer now.

• The unique dataset and forms of data (LSOA, IMD, Mosaic) do require some extended discussions for readers unfamiliar with them, but the better organisation means has also made it more succinct. 

How do the results compare to those of "Outcomes of reablement and their measurement: Findings from an evaluation of English reablement the authors do this in some instances but not all”. Also remember absence of evidence is not evidence of absence services" https://onlinelibrary.wiley.com/doi/full/10.1111/hsc.12814. 

• We read this paper and have incorporated some of its insights as can be seen on new sentences on p.4 (line 100) and at the start of the discussion on p.23

Some discussion about the challenges of the data at hand and the restructuring of services.

• We have incorporated this comment (p.7 and p.21).

New References Included

Wami, Welcome M, Ruth Dundas, Oarabile R Molaodi, Mette Tranter, Alastair H Leyland, and Srinivasa Vittal Katikireddi. 2019. "Assessing the potential utility of commercial ‘big data’for health research: Enhancing small-area deprivation measures with Experian™ Mosaic groups." Health & place 57:238-46.

Moon, Graham, Liz Twigg, Kelvyn Jones, Grant Aitken, and Joanna Taylor. 2019. "The utility of geodemographic indicators in small area estimates of limiting long-term illness." Social Science & Medicine 227:47-55.

Liaaen, J., and K. Vik. 2019. "Becoming an enabler of everyday activity: Health professionals in home care services experiences of working with reablement." Int J Older People Nurs:e12270.

Langeland, E., H. Tuntland, B. Folkestad, O. Forland, F. F. Jacobsen, and I. Kjeken. 2019. "A multicenter investigation of reablement in Norway: a clinical controlled trial." BMC Geriatr 19(1):29.

Jokstad, K., K. Skovdahl, B. T. Landmark, and H. Haukelien. 2019. "Ideal and reality; Community healthcare professionals' experiences of user-involvement in reablement." Health Soc Care Community 27(4):907-16.

Fransham, Mark. 2019. "Income and population dynamics in deprived neighbourhoods: measuring the poverty turnover rate using administrative data." Applied Spatial Analysis and Policy 12(2):275-300.

Doh, Daniel, Ricki Smith, and Paula Gevers. 2019. "Reviewing the reablement approach to caring for older people." Ageing & Society:1-13.

Bodker, M. N., H. Langstrup, and U. Christensen. 2019. "What constitutes 'good care' and 'good carers'? The normative implications of introducing reablement in Danish home care." Health Soc Care Community 27(5):e871-e78.

Bodker, M. N., U. Christensen, and H. Langstrup. 2019. "Home care as reablement or enabling arrangements? An exploration of the precarious dependencies in living with functional decline." Sociol Health Illn 41(7):1358-72.

Beresford, B., E. Mayhew, A. Duarte, R. Faria, H. Weatherly, R. Mann, G. Parker, F. Aspinal, and M. Kanaan. 2019. "Outcomes of reablement and their measurement: Findings from an evaluation of English reablement services." Health Soc Care Community.

Slater, Paul, and Felicity Hasson. 2018. "An evaluation of the reablement service programme on physical ability, care needs and care plan packages." Journal of Integrated Care 26(2):140-49.

Moe, C., and B. S. Brinchmann. 2018. "Tailoring reablement: A grounded theory study of establishing reablement in a community setting in Norway." Health Soc Care Community 26(1):113-21.

Malomo, Fola. 2018. "Why do some coastal communities rise while others decline?" Ocean & Coastal Management 151:92-98.

Jeon, Y. H., L. Clemson, S. L. Naismith, L. Mowszowski, N. McDonagh, M. Mackenzie, C. Dawes, L. Krein, and S. L. Szanton. 2018. "Improving the social health of community-dwelling older people living with dementia through a reablement program." Int Psychogeriatr 30(6):915-20.

Eliassen, M., N. O. Henriksen, and S. Moe. 2018. "Physiotherapy supervision of home trainers in interprofessional reablement teams." J Interprof Care:1-7.

Stepner, Michael. 2017. "MAPTILE: Stata module to map a variable."

Sims-Gould, J., C. E. Tong, L. Wallis-Mayer, and M. C. Ashe. 2017. "Reablement, Reactivation, Rehabilitation and Restorative Interventions With Older Adults in Receipt of Home Care: A Systematic Review." J Am Med Dir Assoc 18(8):653-63.

Poulos, C. J., A. Bayer, L. Beaupre, L. Clare, R. G. Poulos, R. H. Wang, S. Zuidema, and K. S. McGilton. 2017. "A comprehensive approach to reablement in dementia." Alzheimers Dement (N Y) 3(3):450-58.

Pooley, Alison, and Annabel Brown. 2017. "The Almshouse Reimagined: challenging students in creating community."

Moe, A., K. Ingstad, and H. V. Brataas. 2017. "Patient influence in home-based reablement for older persons: qualitative research." Bmc Health Services Research 17(1):736.

Hjelle, K. M., H. Tuntland, O. Forland, and H. Alvsvag. 2017. "Driving forces for home-based reablement; a qualitative study of older adults' experiences." Health Soc Care Community 25(5):1581-89.

---

## [Decision Letter · Decision Letter 1]

13 Mar 2020

PONE-D-19-21555R1

Enabling and constraining successful reablement: Individual and neighbourhood factors

PLOS ONE

Dear Mr Jacobi,

Thank you for submitting your manuscript to PLOS ONE. After careful consideration, we feel that it has merit but does not fully meet PLOS ONE’s publication criteria as it currently stands. Therefore, we invite you to submit a revised version of the manuscript that addresses the points raised during the review process.

There are still some minor suggestions from the reviewer which are appended below. Please respond to those suggestions.

We would appreciate receiving your revised manuscript by Apr 27 2020 11:59PM. To enhance the reproducibility of your results, we recommend that if applicable you deposit your laboratory protocols in protocols.io, where a protocol can be assigned its own identifier (DOI) such that it can be cited independently in the future. For instructions see: http://journals.plos.org/plosone/s/submission-guidelines#loc-laboratory-protocols

We look forward to receiving your revised manuscript.

Kind regards,

Kannan Navaneetham

Academic Editor

PLOS ONE

Reviewers' comments:

Reviewer's Responses to Questions

**Comments to the Author**

1. If the authors have adequately addressed your comments raised in a previous round of review and you feel that this manuscript is now acceptable for publication, you may indicate that here to bypass the “Comments to the Author” section, enter your conflict of interest statement in the “Confidential to Editor” section, and submit your "Accept" recommendation.

Reviewer #1: All comments have been addressed

2. Is the manuscript technically sound, and do the data support the conclusions?

Reviewer #1: (No Response)

3. Has the statistical analysis been performed appropriately and rigorously? 

Reviewer #1: (No Response)

4. Have the authors made all data underlying the findings in their manuscript fully available?

Reviewer #1: (No Response)

5. Is the manuscript presented in an intelligible fashion and written in standard English?

Reviewer #1: (No Response)

6. Review Comments to the Author

Reviewer #1: Thank you for asking me to re-review this manuscript which is much improved by the changes the authors have made. I have a few minor comments that the authors might wish to consider if the editors think these are appropriate.

1. References are needed in a couple of places - p3, line 61 (after 'hospitalisation') and p3, line 64 (after 'others').

2. On page 7 - it is unclear whether 0 or 4 indicates high level of care needs on the SMT (line 172). It is unclear who is responsible for interpreting the SMT (line 174).

3. There are some minor editorial conventions that have not been followed - e.g. sentences starting with a number rather than written in full.

4. To help to the reader's understanding of reablement service provision, it might be useful to include the range of timescales from clients' entry to reablement to 13 weeks post-discharge. This would give an indication of the range of duration of the reablement intervention.

5. I remain unclear why the level of social care need in hours were grouped as they are - is there a clinical/substantive significant difference between people requiring 9 hours and those requiring 10 hours of input?

7. PLOS authors have the option to publish the peer review history of their article (what does this mean?). If published, this will include your full peer review and any attached files.

Reviewer #1: No

---

## [Author Response · Author response to Decision Letter 1]

27 Jul 2020

We thank the reviewer for the excellent comments, and we are delighted to read that the changes after the first round of reviews have strongly improved the paper. In the following paragraphs, the leading sentence is taken from the reviewer’s comments and the bullet points correspond to the changes that we have implemented and our responses. 

Reviewer #1: Thank you for asking me to re-review this manuscript which is much improved by the changes the authors have made. I have a few minor comments that the authors might wish to consider if the editors think these are appropriate.

• We thank the reviewer for the helpful and positive comments. We have incorporated all of the suggested changes.

1. References are needed in a couple of places - p3, line 61 (after 'hospitalisation') and p3, line 64 (after 'others').

• We have added appropriate references on page 3 lines 64 and 66. As part of this, we are including a new and very recent article:

o Hu FW, Huang YT, Lin HS, Chen CH, Chen MJ, Chang CM. Effectiveness of a simplified reablement program to minimize functional decline in hospitalized older patients. Geriatrics & Gerontology International. 2020;20(5):436-42.

• The other three citations are:

o Cochrane A, Furlong M, McGilloway S, Molloy DW, Stevenson M, Donnelly M. Time-limited home-care reablement services for maintaining and improving the functional independence of older adults. Cochrane Database Syst Rev. 2016;10:CD010825.

o Legg L, Gladman J, Drummond A, Davidson A. A systematic review of the evidence on home care reablement services. Clin Rehabil. 2016;30(8):741-9.

o Lewin GF, Alfonso HS, Alan JJ. Evidence for the long term cost effectiveness of home care reablement programs. Clin Interv Aging. 2013;8:1273-81.

2. On page 7 - it is unclear whether 0 or 4 indicates high level of care needs on the SMT (line 172). It is unclear who is responsible for interpreting the SMT (line 174).

• This is a good suggestion in terms of the clarity of the reablement Service Assessment Tool (SMT) and we have thus expanded the explanations of the care-need scoring mechanism on page 7 line 174. The ordinal SMT is scored from 0 “full dependence or inability” to 4 “full independence”.

• We have expanded the explanations of the SMT: “Individual clients were then assigned a reablement package tailored to meet their individual needs as ascertained by the private care company’s use of the SMT.” (page 7 lines 351-353).

3. There are some minor editorial conventions that have not been followed - e.g. sentences starting with a number rather than written in full.

• We have reworded section 3.1 (case selection), especially between lines 191 and 207. The phrasing is now more polished as we no longer start sentences with a number. We have checked the paper for other editorial mistakes too.

4. To help to the reader's understanding of reablement service provision, it might be useful to include the range of timescales from clients' entry to reablement to 13 weeks post-discharge. This would give an indication of the range of duration of the reablement intervention.

• Yes, this is a helpful comment. We discussed the timescale of the reablement intervention in the initially submitted version of the manuscript, but later removed it as it is not the main part of the analysis. However, we fully agree that the duration or rather length of reablement episodes should be stated and have done so in lines 206 and 207 on page 9 “The selected reablement episodes had an average duration of 35 days and a median duration 38 days.”

5. I remain unclear why the level of social care need in hours were grouped as they are - is there a clinical/substantive significant difference between people requiring 9 hours and those requiring 10 hours of input?

• The categorical groupings of social care need in hours were determined by the occurrence of clusters of typical scores (e.g. the specific scores of 3.5 hours, 7 hours, and 8.75 hours covered about a fairly large number of episodes) as based on the SMT. Our chosen categorization in steps of three hours of care need covered this distribution fairly accurately – while a truly nominal/linear one would not be justified. A similar categorization was also employed by Essex County Council, the data provider, when they had an initial look at the data for their own purposes. As we are dealing with administrative data, a more detailed form of measurement is not available.

• However, we acknowledge this set of limitations in section 5.1 and state that “moreover, in order to understand the significant factors that have been identified with more certainty and accuracy, further research is required into the specific nature of individual reablement programmes and the details of the underlying disabilities of reablement clients.”

• Despite these practical and distributional reasons for a categorical specification, we have still tested all models with a linear one (care need in hours as a continuous predictor variable) and the results were virtually identical and remained statistically significant. The coefficients of the covariates only showed negligible difference at the third decimal point. One could argue that our categorical categorization of care need is -statistically speaking - more conservative than the continuous specification.

---

## [Editor Report · Decision Letter 2]

28 Jul 2020

Enabling and constraining successful reablement: Individual and neighbourhood factors

PONE-D-19-21555R2

Dear Dr. Jacobi,

We’re pleased to inform you that your manuscript has been judged scientifically suitable for publication and will be formally accepted for publication once it meets all outstanding technical requirements.

Kind regards,

Kannan Navaneetham, PhD

Academic Editor

PLOS ONE
---

## [Editor Report · Acceptance letter]

27 Aug 2020

PONE-D-19-21555R2 

Enabling and constraining successful reablement: Individual and neighbourhood factors 

Dear Dr. Jacobi:

I'm pleased to inform you that your manuscript has been deemed suitable for publication in PLOS ONE. Congratulations! Your manuscript is now with our production department. 

Kind regards, 

on behalf of

Professor Kannan Navaneetham 

Academic Editor

PLOS ONE